# MULTI-RATE VAE: TRAIN ONCE, GET THE FULL RATE-DISTORTION CURVE

**Juhan Bae**[1,2]*, **Michael R. Zhang**[1,2], **Michael Ruan**[1], **Eric Wang**[1], **So Hasegawa**[3],
**Jimmy Ba**[1,2], **Roger Grosse**[1,2,4]
[1]University of Toronto, [2]Vector Institute, [3]Fujitsu Limited, [4]Anthropic

## ABSTRACT

Variational autoencoders (VAEs) are powerful tools for learning latent representations of data used in a wide range of applications. In practice, VAEs usually require multiple training rounds to choose the amount of information the latent variable should retain. This trade-off between the reconstruction error (distortion) and the KL divergence (rate) is typically parameterized by a hyperparameter $\beta$. In this paper, we introduce Multi-Rate VAE (MR-VAE), a computationally efficient framework for learning optimal parameters corresponding to various $\beta$ in a single training run. The key idea is to explicitly formulate a response function using hypernetworks that maps $\beta$ to the optimal parameters. MR-VAEs construct a compact response hypernetwork where the pre-activations are conditionally gated based on $\beta$. We justify the proposed architecture by analyzing linear VAEs and showing that it can represent response functions exactly for linear VAEs. With the learned hypernetwork, MR-VAEs can construct the rate-distortion curve without additional training and can be deployed with significantly less hyperparameter tuning. Empirically, our approach is competitive and often exceeds the performance of multiple $\beta$-VAEs training with minimal computation and memory overheads.

## 1 INTRODUCTION

Deep latent variable models sample latent factors from a prior distribution and convert them to realistic data points using neural networks. However, computing the model parameters via maximum likelihood estimation is challenging due to the need to marginalize the latent factors, which is intractable. *Variational Autoencoders (VAEs)* (Kingma & Welling, 2013; Rezende et al., 2014) formulate a tractable lower bound for the log-likelihood and enable optimization of deep latent variable models by reparameterization of the *Evidence Lower Bound (ELBO)* (Jordan et al., 1999). VAEs have been applied in many different contexts, including text generation (Bowman et al., 2015), data augmentation generation (Norouzi et al., 2020), anomaly detection (An & Cho, 2015; Park et al., 2022), future frame prediction (Castrejon et al., 2019), image segmentation (Kohl et al., 2018), and music generation (Roberts et al., 2018).

In practice, VAEs are typically trained with the $\beta$-VAE objective (Higgins et al., 2016) which balances the reconstruction error (*distortion*) and the KL divergence term (*rate*):

$$\mathcal{L}_\beta(\boldsymbol{\phi}, \boldsymbol{\theta}) = \underbrace{\mathbb{E}_{p_d(\mathbf{x})}[\mathbb{E}_{q_\phi(\mathbf{z}|\mathbf{x})}[-\log p_\theta(\mathbf{x}|\mathbf{z})]]}_{\text{Distortion}} + \beta \underbrace{\mathbb{E}_{p_d(\mathbf{x})}[D_{\text{KL}}(q_\phi(\mathbf{z}|\mathbf{x}), p(\mathbf{z}))]}_{\text{Rate}}, \tag{1}$$

where $p_\theta(\mathbf{x}|\mathbf{z})$ models the process that generates the data $\mathbf{x}$ given the latent variable $\mathbf{z}$ (the "decoder") and $q_\phi(\mathbf{z}|\mathbf{x})$ is the variational distribution (the "encoder"), parameterized by $\boldsymbol{\theta} \in \mathbb{R}^m$ and $\boldsymbol{\phi} \in \mathbb{R}^p$, respectively. Here, $p(\mathbf{z})$ is a prior on the latent variables, $p_d(\mathbf{x})$ is the data distribution, and $\beta > 0$ is the weight on the KL term that trades off between rate and distortion.

On the one hand, models with low distortions can reconstruct data points with high quality but may generate unrealistic data points due to large discrepancies between variational distributions and priors (Alemi et al., 2018). On the other hand, models with low rates have variational distributions close to the prior but may not have encoded enough useful information to reconstruct the data. Hence,

---

*Correspondence to `jbae@cs.toronto.edu`.

| Methods | Parameters | Training Time (s) | RD Curve |
|---|---|---|---|
| **Standard VAE** | $18.81 \times 10^4$ | 4,832 | ✗ |
| **$\beta$-VAE** | $18.81 \times 10^4$ | 4,836 | ✗ |
| **$\beta$-VAEs** (+8 runs) | $15.04 \times 10^5$ | 38,492 | ▲ |
| **$\beta$-VAEs** (+ KL Annealing) | $15.04 \times 10^5$ | 38,640 | ▲ |
| **MR-VAEs** (ours) | $18.84 \times 10^4$ | 5,040 | ✓ |

**Figure 1:** $\beta$-VAEs require multiple runs of training with different KL weights $\beta$ to visualize parts of the rate-distortion curve (Pareto frontier). Our proposed Multi-Rate VAEs (MR-VAEs) can learn the full continuous rate-distortion curve in a single run with small memory and computational overhead.

the KL weight $\beta$ plays an important role in training VAEs and requires careful tuning for various applications (Kohl et al., 2018; Castrejon et al., 2019; Pong et al., 2019). The KL weighting term also has a close connection to disentanglement quality (Higgins et al., 2016; Burgess et al., 2018; Nakagawa et al., 2021), generalization ability (Kumar & Poole, 2020; Bozkurt et al., 2021), data compression (Zhou et al., 2018; Huang et al., 2020), and posterior collapse (Lucas et al., 2019; Dai et al., 2020; Wang & Ziyin, 2022).

By training multiple VAEs with different values of $\beta$, we can obtain different points on a rate-distortion curve (Pareto frontier) from information theory (Alemi et al., 2018). Unfortunately, as rate-distortion curves depend on both the dataset and architecture, practitioners generally need to tune $\beta$ for each individual task. In this work, we introduce a modified VAE framework that does not require hyperparameter tuning on $\beta$ and can learn multiple VAEs with different rates in a single training run. Hence, we call our approach *Multi-Rate VAE (MR-VAE)*.

We first formulate *response functions* $\phi^\star(\beta)$ and $\theta^\star(\beta)$ (Gibbons et al., 1992) which map the KL weight $\beta$ to the optimal encoder and decoder parameters trained with such $\beta$. Next, we explicitly construct response functions $\phi_\psi(\beta)$ and $\theta_\psi(\beta)$ using hypernetworks (Ha et al., 2016), where $\psi \in \mathbb{R}^h$ denotes hypernetwork parameters. Unlike the original VAE framework, which requires retraining the network to find optimal parameters for some particular $\beta$, response hypernetworks can directly learn this mapping and do not require further retraining.

While there is a lot of freedom in designing the response hypernetwork, we propose a hypernetwork parameterization that is memory and cost-efficient yet flexible enough to represent the optimal parameters over a wide range of KL weights. Specifically, in each layer of a VAE, our MR-VAE architecture applies an affine transformation to $\log \beta$ and uses it to scale the pre-activation. We justify the proposed architecture by analyzing linear VAEs and showing that the MR-VAE architecture can represent the response functions on this simplified model. We further propose a modified objective analogous to Self-Tuning Networks (MacKay et al., 2019; Bae & Grosse, 2020) to optimize response hypernetworks instead of the standard encoder and decoder parameters.

Empirically, we trained MR-VAEs to learn rate-distortion curves for image and text reconstruction tasks over a wide range of architectures. Across all tasks and architectures, MR-VAEs found competitive or even improved rate-distortion curves compared to the baseline method of retraining the network multiple times with different KL weights. We show a comparison between $\beta$-VAE (with and without KL annealing (Bowman et al., 2015)) and MR-VAE with ResNet-based encoders and decoders (He et al., 2016) in Figure 1. MR-VAEs can learn multiple optimal parameters corresponding to various KL weights in a single training run. Moreover, MR-VAEs do not require KL weight schedules and can be deployed without significant hyperparameter tuning.

Our framework is general and can be extended to various existing VAE models. We demonstrate this flexibility by applying MR-VAEs to $\beta$-TCVAEs (Chen et al., 2018), where we trade-off the reconstruction error and total correlation instead of the reconstruction error and rate. We show that MR-VAEs can be used to evaluate the disentanglement quality over a wide range of $\beta$ values without having to train $\beta$-TCVAEs multiple times.

## 2 BACKGROUND

### 2.1 ($\beta$-) VARIATIONAL AUTOENCODERS

Variational Autoencoders jointly optimize encoder parameters $\phi$ and decoder parameters $\theta$ to minimize the $\beta$-VAE objective defined in Eqn. 1. While the standard ELBO sets the KL weight to 1,

VAEs are commonly trained with varying KL weight $\beta$. As the KL weight $\beta$ approaches 0, the VAE objective resembles the deterministic autoencoder, putting more emphasis on minimizing the reconstruction error. When the KL weight is large, the objective prioritizes minimizing the KL divergence term, and the variational distribution may collapse to the prior.

It is possible to interpret VAEs from an information-theoretic framework (Alemi et al., 2018), where the $\beta$-VAE objective is a special case of the information-bottleneck (IB) method (Tishby et al., 2000; Alemi et al., 2016). In this perspective, the decoder $p_{\boldsymbol{\theta}}(\mathbf{x}|\mathbf{z})$ serves as a lower bound for the mutual information $I_q(\mathbf{x}; \mathbf{z})$ and the rate upper bounds the mutual information. The relationship can be summarized as follows:

$$H - D \leq I_q(\mathbf{x}; \mathbf{z}) \leq R, \tag{2}$$

where $D$ is the distortion, $H$ is the data entropy that measures the complexity of the dataset, and $R$ is the rate. Note that we adopt terminologies "rate" and "distortion" from rate-distortion theory (Thomas & Joy, 2006; Alemi et al., 2018; Bozkurt et al., 2021), which aims to minimize the rate under some constraint on distortion. The $\beta$-VAE objective in Eqn. 1 can be seen as a Lagrangian relaxation of the rate-distortion objective:

$$\mathcal{L}_{\beta}(\boldsymbol{\phi}, \boldsymbol{\theta}) = D + \beta R. \tag{3}$$

In this view, training multiple VAEs with different values of $\beta$ corresponds to different points on the rate-distortion curve and distinguishes models that do not utilize latent variables and models that make large use of latent variables (Phuong et al., 2018). We refer readers to Alemi et al. (2018) for a more detailed discussion of the information-theoretic framework.

## 2.2 Linear VAEs

The linear VAE is a simple model where both the encoder and decoder are constrained to be affine transformations. More formally, we consider the following problem from Lucas et al. (2019):

$$\begin{aligned} p_{\boldsymbol{\theta}}(\mathbf{x}|\mathbf{z}) &= \mathcal{N}(\mathbf{D}\mathbf{z} + \boldsymbol{\mu}, \sigma^2 \mathbf{I}) \\ q_{\boldsymbol{\phi}}(\mathbf{z}|\mathbf{x}) &= \mathcal{N}(\mathbf{E}(\mathbf{x} - \boldsymbol{\mu}), \mathbf{C}), \end{aligned} \tag{4}$$

where $\mathbf{C}$ is a diagonal covariance matrix that is shared across all data points, $\mathbf{E}$ is the encoder weight, and $\mathbf{D}$ is the decoder weight.

While the analytic solutions for deep latent models typically do not exist, the linear VAE has analytic solutions for optimal parameters and allows us to reason about various phenomena in training VAEs. For example, Dai et al. (2018) show the relationship of linear VAE, probabilistic PCA (pPCA) (Tipping & Bishop, 1999), and robust PCA (Candès et al., 2011; Chandrasekaran et al., 2011) and analyze the local minima smoothing effects of VAEs. Similarly, Lucas et al. (2019) and Wang & Ziyin (2022) use linear VAEs to investigate the cause of the posterior collapse (Razavi et al., 2019).

## 2.3 Response Functions

Response (rational reaction) functions (Gibbons et al., 1992; Lorraine & Duvenaud, 2018) in neural networks map a set of hyperparameters to the optimal parameters trained with such hyperparameters. In the case of $\beta$-VAEs, response functions map the KL weight $\beta$ to optimal encoder and decoder parameters that minimize the $\beta$-VAE objective:

$$\begin{aligned} \boldsymbol{\phi}^{\star}(\beta) &= \underset{\boldsymbol{\phi} \in \mathbb{R}^p}{\arg\min} \, \mathcal{L}_{\beta}(\boldsymbol{\phi}, \boldsymbol{\theta}) \\ \boldsymbol{\theta}^{\star}(\beta) &= \underset{\boldsymbol{\theta} \in \mathbb{R}^m}{\arg\min} \, \mathcal{L}_{\beta}(\boldsymbol{\phi}, \boldsymbol{\theta}). \end{aligned} \tag{5}$$

Approximation of response functions has various applications in machine learning, including hyper-parameter optimization (Lorraine & Duvenaud, 2018), game theory (Fiez et al., 2019; Wang et al., 2019), and influence estimation (Bae et al., 2022).

## 3 Methods

In this section, we introduce Multi-Rate VAEs (MR-VAEs), an approach for directly modeling VAE response functions with hypernetworks. We first formulate memory and cost-efficient response

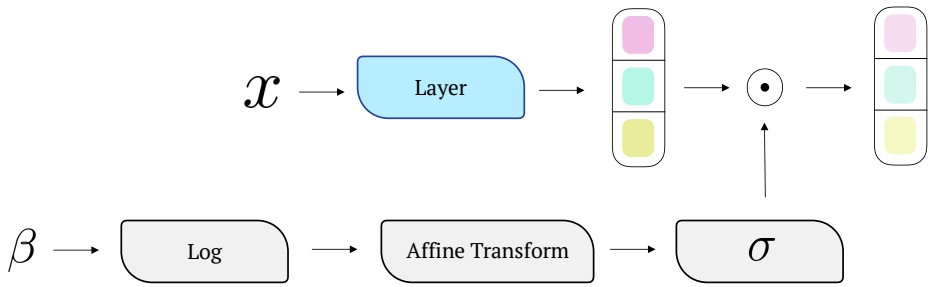

**Figure 2:** The MR-VAE architecture scales the pre-activations for each layer of the base VAE. This scaling term is generated with an affine transformation on the log KL weight followed by an activation function.

hypernetworks $\phi_{\psi}(\beta)$ and $\theta_{\psi}(\beta)$ and justify the proposed parameterization by showing that they recover exact response functions for linear VAEs. Then, we propose a modified training objective for MR-VAEs and formally describe an algorithm for learning the entire rate-distortion curve in a single training run.

### 3.1 RESPONSE HYPERNETWORKS FOR VAES

We propose explicitly modeling response functions with hypernetworks to construct the rate-distortion curve, as opposed to training multiple VAEs with different KL weights $\beta$. The formulation of response hypernetworks, shown in Figure 3, is advantageous as we can infer the optimal parameters for some $\beta$ by simply changing the inputs to the hypernetwork. As response functions map a scalar to a high-dimensional parameter space $\mathbb{R}^{m+p}$, there are many possible architectural designs for such response hypernetworks.

Our Multi-Rate VAEs (MR-VAEs) construct a compact approximation of response functions by scaling each layer's weights and bias by a function of the KL

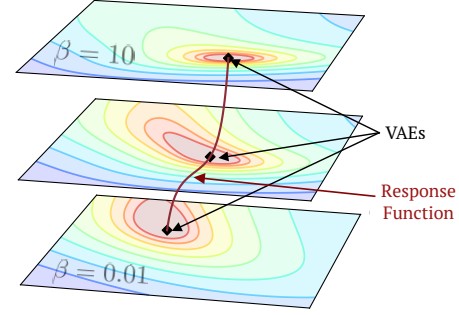

**Figure 3:** Instead of training several VAEs for each desired KL weight $\beta$, MR-VAEs learn the response functions with a hypernetwork in a single training run.

weight. More formally, consider the $i$-th layer of a VAE, whose weight and bias are expressed as $\mathbf{W}^{(i)} \in \mathbb{R}^{m_{i+1} \times m_i}$ and $\mathbf{b}^{(i)} \in \mathbb{R}^{m_{i+1}}$, respectively. We directly model response functions with hypernetworks $\psi \in \mathbb{R}^h$ by parameterizing each weight and bias as follows[1]:

$$\mathbf{W}^{(i)}_{\psi}(\beta) = \sigma^{(i)} \left( \mathbf{w}^{(i)}_{\text{hyper}} \log(\beta) + \mathbf{b}^{(i)}_{\text{hyper}} \right) \odot_{\text{row}} \mathbf{W}^{(i)}_{\text{base}}$$
$$\mathbf{b}^{(i)}_{\psi}(\beta) = \sigma^{(i)} \left( \mathbf{w}^{(i)}_{\text{hyper}} \log(\beta) + \mathbf{b}^{(i)}_{\text{hyper}} \right) \odot \mathbf{b}^{(i)}_{\text{base}},$$

(6)

where $\odot$ and $\odot_{\text{row}}$ indicate elementwise multiplication and row-wise rescaling. Here, $\mathbf{w}^{(i)}_{\text{hyper}}, \mathbf{b}^{(i)}_{\text{hyper}} \in \mathbb{R}^{m_{i+1}}$ are vectors that are used in applying an affine transformation to the log KL weight. We further define the elementwise activation function $\sigma^{(i)}(\cdot)$ as:

$$\sigma^{(i)}(x) = \begin{cases} \dfrac{1}{1 + e^{-x}} & \text{if } i\text{-th layer belongs to encoder } \boldsymbol{\theta}. \\ (\text{ReLU}(1 - \exp(x)))^{1/2} & \text{if } i\text{-th layer belongs to decoder } \boldsymbol{\phi}. \end{cases}$$

(7)

This specific choice of activation functions recovers the exact response function for linear VAEs, which we elaborate on in Section 3.2. We further note that the above activation functions scale the base weight and bias between 0 and 1. In Appendix D, we show that the choice of activation function does not significantly change the performance of MR-VAEs and we can use sigmoid gatings for both encoders and decoders in practice.

Observe that the hypernetwork parameterization in Eqn. 6 is both memory and cost-efficient: it needs $2m_{i+1}$ additional parameters to represent the weight and bias and requires 2 additional elementwise

---

[1]We show the hypernetwork parameterization for convolution layers in Appendix B.

multiplications per layer in the forward pass. Given activations $\mathbf{a}^{(i-1)} \in \mathbb{R}^{m_i}$ from the previous layer, the resulting pre-activations $\mathbf{s}^{(i)}$ can alternatively be expressed as:

$$\mathbf{s}^{(i)} = \mathbf{W}_{\boldsymbol{\psi}}^{(i)}(\beta)\mathbf{a}^{(i-1)} + \mathbf{b}_{\boldsymbol{\psi}}^{(i)}(\beta) \tag{8}$$

$$= \sigma^{(i)}\left(\mathbf{w}_{\text{hyper}}^{(i)}\log(\beta) + \mathbf{b}_{\text{hyper}}^{(i)}\right) \odot \left(\mathbf{W}_{\text{base}}^{(i)}\mathbf{a}^{(i-1)} + \mathbf{b}_{\text{base}}^{(i)}\right). \tag{9}$$

The architecture for MR-VAEs is illustrated in Figure 2. This response hypernetwork is straightforward to implement in popular deep learning frameworks (e.g., `PyTorch` (Paszke et al., 2019) and `Jax` (Bradbury et al., 2018)) by replacing existing modules with pre-activations gated modules. We provide sample `PyTorch` code in Appendix B.2.

## 3.2 EXACT RESPONSE FUNCTIONS FOR LINEAR VAEs

Here, we justify the proposed hypernetwork parameterization introduced in Section 3.1 by analyzing the exact response functions for linear VAEs (Lucas et al., 2019). We show that MR-VAEs can precisely express the response functions for this simplified model. Consider the following problem (analogous to the setup introduced in Section 2.2):

$$p_{\boldsymbol{\theta}}(\mathbf{x}|\mathbf{z}) = \mathcal{N}(\mathbf{D}^{(2)}\mathbf{D}^{(1)}\mathbf{z} + \boldsymbol{\mu}, \sigma^2\mathbf{I})$$
$$q_{\boldsymbol{\phi}}(\mathbf{z}|\mathbf{x}) = \mathcal{N}(\mathbf{E}^{(2)}\mathbf{E}^{(1)}(\mathbf{x} - \boldsymbol{\mu}), \mathbf{C}^{(2)}\mathbf{C}^{(1)}). \tag{10}$$

Compared to the linear VAE setup in Eqn. 4, we represent the encoder weight, diagonal covariance, and decoder weight as the product of two matrices for compatibility with our framework, where $\mathbf{E}^{(1)}, \mathbf{C}^{(1)}$, and $\mathbf{D}^{(1)}$ are all square matrices. This decomposition can be understood as constructing two-layer linear networks for both the encoder and decoder.

We apply the MR-VAE formulation (shown in Eqn. 6) to all parameters and construct a response hypernetwork for each encoder weight, diagonal covariance, and decoder weight. The following theorem shows that the MR-VAE architecture is capable of representing the response functions for linear VAEs.

**Theorem 1.** *Consider the linear VAE model introduced in Eqn. 10. Let the encoder weight, diagonal covariance matrix, and decoder weight be parameterized as $\mathbf{E}_{\boldsymbol{\psi}}(\beta) = \mathbf{E}_{\boldsymbol{\psi}}^{(2)}(\beta)\mathbf{E}_{\boldsymbol{\psi}}^{(1)}(\beta)$, $\mathbf{C}_{\boldsymbol{\psi}}(\beta) = \mathbf{C}_{\boldsymbol{\psi}}^{(2)}(\beta)\mathbf{C}_{\boldsymbol{\psi}}^{(1)}(\beta)$, and $\mathbf{D}_{\boldsymbol{\psi}}(\beta) = \mathbf{D}_{\boldsymbol{\psi}}^{(2)}(\beta)\mathbf{D}_{\boldsymbol{\psi}}^{(1)}(\beta)$, respectively. Then, there exist hypernetwork parameters $\boldsymbol{\psi}^{\star} \in \mathbb{R}^h$ such that $\mathbf{E}_{\boldsymbol{\psi}^{\star}}(\beta), \mathbf{C}_{\boldsymbol{\psi}^{\star}}(\beta)$, and $\mathbf{D}_{\boldsymbol{\psi}^{\star}}(\beta)$ are the exact response functions for linear VAEs.*

The proof is shown in Appendix A.2. Note that we use a separate activation function (described in Eqn. 7) for encoder and decoder parameters to exactly represent the response functions. Motivated by the analysis presented in this section, MR-VAEs use the same hypernetwork parameterization for deep VAEs.

## 3.3 TRAINING OBJECTIVE FOR RESPONSE HYPERNETWORK

To learn the optimal parameters for various ranges of the KL weight $\beta$, we propose to use an objective analogous to the Self-Tuning Networks (STN) objective (Lorraine & Duvenaud, 2018; MacKay et al., 2019; Bae & Grosse, 2020). The hypernetwork parameters $\boldsymbol{\psi}$ are optimized to minimize the following objective:

$$\boldsymbol{\psi}^{\star} = \underset{\boldsymbol{\psi} \in \mathbb{R}^h}{\arg\min} \, \mathcal{Q}(\boldsymbol{\psi}) = \underset{\boldsymbol{\psi} \in \mathbb{R}^h}{\arg\min} \, \mathbb{E}_{\eta \sim \mathcal{U}[\log(a), \log(b)]}\left[\mathcal{L}_{\exp(\eta)}(\boldsymbol{\phi}_{\boldsymbol{\psi}}(\exp(\eta)), \boldsymbol{\theta}_{\boldsymbol{\psi}}(\exp(\eta)))\right], \tag{11}$$

where $\mathcal{U}[\log(a), \log(b)]$ is an uniform distribution with range $\log(a)$ and $\log(b)$. The proposed objective encourages response hypernetworks to learn optimal parameters for all KL weights $\beta$ in the range between $a$ and $b$. Unlike the STN objective, we sample the hypernetwork inputs from a uniform distribution with a fixed range (instead of a Gaussian distribution with a learnable covariance matrix) as we are interested in learning the global response function (instead of the local response function).

## 3.4 TRAINING ALGORITHM

The complete algorithm for MR-VAE is summarized in Alg. 1. When training MR-VAEs, we approximate the objective in Eqn. 11 by sampling $\eta$ in each gradient descent iteration.

---

**Algorithm 1** Multi-Rate Variational Autoencoders (MR-VAEs)

---

**Require:** $\psi$ (Hypernetwork parameters), $\eta$ (learning rate), $(a, b)$ (sample range)
**while** not converged **do**
$\quad \mathcal{B} \sim \mathcal{D}_{\text{train}}$ $\qquad\qquad\qquad\qquad\qquad\qquad\qquad$ ▷ Sample a mini-batch
$\quad \boldsymbol{\eta} \sim \mathcal{U}(\log(a), \log(b))$ $\qquad\qquad\qquad\qquad$ ▷ Sample inputs to the hypernetwork
$\quad \mathcal{Q}(\psi) := \mathcal{L}_{\exp(\boldsymbol{\eta})}(\phi_\psi(\exp(\boldsymbol{\eta})), \boldsymbol{\theta}_\psi(\exp(\boldsymbol{\eta})); \mathcal{B})$ $\qquad$ ▷ Compute the MR-VAE objective
$\quad \psi \leftarrow \psi - \eta \nabla_\psi \mathcal{Q}(\psi)$ $\qquad\qquad\qquad\qquad\qquad$ ▷ Update hypernetwork parameters
**end while**

---

**Normalizations.** As it is widely considered beneficial for optimization to normalize the inputs and activations (LeCun et al., 2012; Montavon & Müller, 2012; Ioffe & Szegedy, 2015; Bae & Grosse, 2020), we standardize the inputs to the hypernetworks (after $\log$ transform) to have a mean of 0 and standard deviation of 1. Since the inputs are sampled from a fixed distribution $\mathcal{U}[\log(a), \log(b)]$, we can apply a fixed transformation based on the sample range $[a, b]$ specified before training.

**Memory and Computation Costs.** As detailed in Section 3.1, MR-VAE architectures introduce 2 additional vectors for each layer of the base neural network. During the forward pass, MR-VAEs require 2 additional elementwise operations in each layer. Across all our image reconstruction experiments, MR-VAEs require at most 5% increase in parameters and wall-clock time compared to $\beta$-VAEs.

**Hyperparameters.** While MR-VAE does not require hyperparameter tuning on the KL weight, it needs two hyperparameters $a$ and $b$ that define the sample range for the KL weight. However, we show that MR-VAEs are robust to the choices of these hyperparameters in Section 5.3 and we use a fixed value $a = 0.01$ and $b = 10$ for our image and text reconstruction experiments.

## 4 RELATED WORKS

**Rate-distortion with VAEs.** Alemi et al. (2018) and Huang et al. (2020) advocate for reporting points along the rate-distortion curve rather than just the ELBO objective to better characterize the representation quality of the latent code. For instance, a powerful decoder can ignore the latent code and still obtain a high marginal likelihood as observed by Bowman et al. (2015) and Chen et al. (2016). Alemi et al. (2018) show that this problem can be alleviated by choosing $\beta < 1$, but this approach still requires training a VAE for each desired information theoretic trade-off. Yang et al. (2020) study the use of modulating activations of autoencoders for image compression, but unlike MR-VAEs, they do not learn a generative model and require learning a separate network for each target rate. Concurrently, Collins et al. (2022) directly provide $\beta$ as inputs to VAEs to learn the rate-distortion trade-off for particle physics discoveries.

**Calibrated Decoders.** Instead of using a fixed decoder variance, calibrated decoders (Lucas et al., 2019; Rybkin et al., 2020) update the decoder variance during training and do not require tuning the KL weight $\beta$. In the case where Gaussian decoders are used, the decoder variance $\sigma^2$ is equivalent to setting $\beta = 2\sigma^2$. Calibrated decoders may be desirable for applications such as density modeling as it is trained with the standard ELBO objective ($\beta = 1$). However, many applications of VAEs involve directly tuning $\beta$. In addition, calibrated decoders cannot construct a rate-distortion curve and may not be appealing when the aim is to get better disentanglement. MR-VAEs can be directly applied in both of these settings.

**Multiplicative Interactions.** Multiplicative interactions have a long history in neural networks. Earlier works include LSTM cells (Hochreiter & Schmidhuber, 1997), which use gating mechanisms to model long-term memories, and Mixture of Experts (Jacobs et al., 1991), which selects learners with a softmax gating. The FiLM network (Perez et al., 2018) and conditional batch normalization (Perez et al., 2017; Brock et al., 2018) further extend the gating mechanism by scaling and shifting the pre-activations. Jayakumar et al. (2020) analyze the expressive power of multiplicative interactions within a network. Our work uses multiplicative interactions to learn the response function for $\beta$-VAEs.

**Response Hypernetworks.** Our use of hypernetworks for learning best response functions is related to Self-Tuning Networks (STNs) (MacKay et al., 2019; Bae & Grosse, 2020) and Pareto Front Hypernetworks (PFHs) (Navon et al., 2020). Lorraine & Duvenaud (2018) first constructed response hypernetworks for bilevel optimization (Colson et al., 2007) and optimized both parameters and hyperparameters in a single training run. To support high dimensional hypernetwork inputs, STNs use structured response hypernetworks similar to MR-VAEs, whereas PFHs use chunking that iteratively generates network parameters. In contrast to these works, the architecture of MR-VAEs is specifically designed for learning $\beta$-VAE's response functions and requires significantly less computation and memory.

## 5 EXPERIMENTS

Our experiments investigate the following questions: (1) Can MR-VAEs learn the optimal response functions for linear VAEs? (2) Does our method scale to modern-size $\beta$-VAEs? (3) How sensitive are MR-VAEs to hyperparameters? (4) Can MR-VAEs be applied to other VAE models such as $\beta$-TCVAEs (Chen et al., 2018)?

We trained MR-VAEs to learn VAEs with multiple rates in a single training run on image and text reconstruction tasks using several network architectures, including convolutional-based VAEs, autoregressive VAEs, and hierarchical VAEs. In addition, we applied MR-VAEs to the $\beta$-TCVAE objective to investigate if our method can be extended to other models and help find VAEs with improved disentangled representations. Experiment details and additional experiments, including ablation studies, are provided in Appendix C and Appendix D, respectively.

### 5.1 HOW DO MR-VAEs PERFORM ON LINEAR VAEs?

We first considered two-layer linear VAE models (introduced in Section 3.2) to validate our theoretical findings. We trained MR-VAEs on the MNIST dataset (Deng, 2012) by sampling the KL weighting term $\beta$ from 0.01 to 10. Then, we trained 10 separate two-layer linear $\beta$-VAEs, each with different $\beta$ values. The rate-distortion curve for MR-VAEs and $\beta$-VAEs are shown in Figure 4. We observed that the rate-distortion curve generated with MR-VAEs closely aligns with the rate-distortion curve generated with individual retraining. Furthermore, by explicitly formulating response functions, MR-VAEs does not require retraining a VAE for each $\beta$.

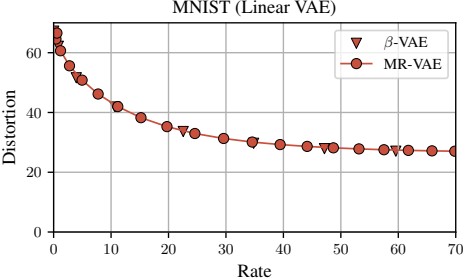

**Figure 4:** A comparison of MR-VAEs and $\beta$-VAEs on linear VAEs. MR-VAEs learn the optimal rate-distortion curve in a single training run.

### 5.2 CAN MR-VAEs SCALE TO MODERN-SIZE ARCHITECTURES?

We trained convolution and ResNet-based architectures on binary static MNIST (Larochelle & Murray, 2011), Omniglot (Lake et al., 2015), CIFAR-10 (Krizhevsky et al., 2009), SVHN (Netzer et al., 2011), and CelebA64 (Liu et al., 2015) datasets, following the experimental set-up from Chadebec et al. (2022). We also trained NVAEs on binary static MNIST, Omniglot, and CelebA datasets using the same experimental set-up from Vahdat & Kautz (2020). Lastly, we trained autoregressive LSTM VAEs on the Yahoo dataset with the set-up from He et al. (2019).

We sampled the KL weighting term from 0.01 to 10 for both MR-VAEs and $\beta$-VAEs. Note that the training for $\beta$-VAEs was repeated 10 times (4 times for NVAEs) with log uniformly spaced $\beta$ to estimate a rate-distortion curve. We compared MR-VAEs with individually trained $\beta$-VAEs and show test rate-distortion curves in Figure 5. Note that traditional VAEs typically require scheduling the KL weight to avoid posterior collapse (Bowman et al., 2015; Fu et al., 2019; Lucas et al., 2019) and we train $\beta$-VAEs with both a constant and KL annealing schedule. Across all tasks, MR-VAEs achieved competitive rate-distortion curves with the independently trained VAEs and even improved performance on the MNIST, Omniglot, and CelebA datasets. Note that we focus on the visualization of the rate-distortion curve as advocated for by Alemi et al. (2016); Huang et al. (2020).

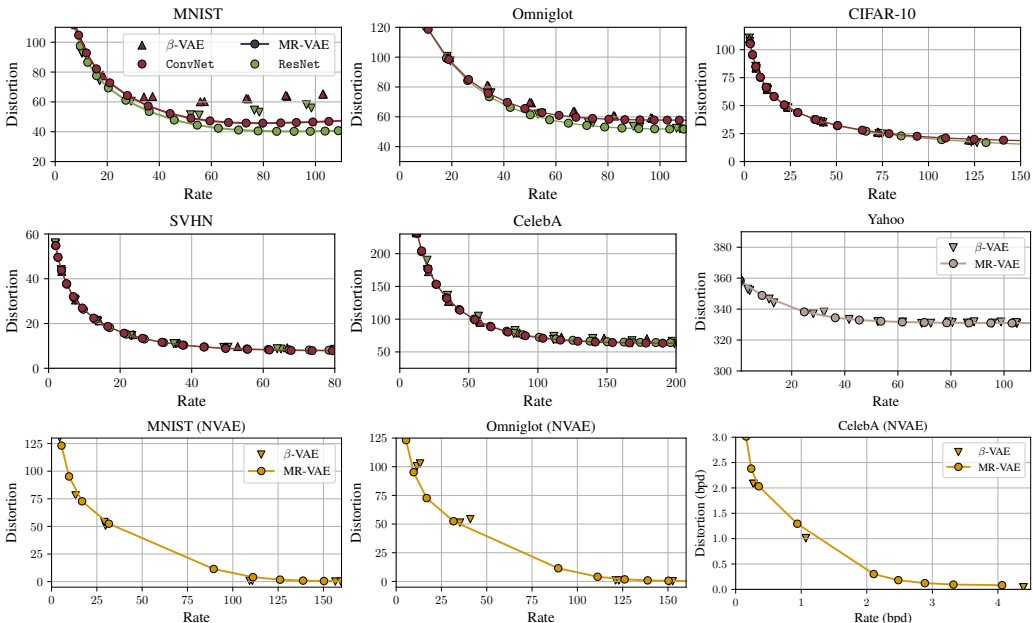

**Figure 5:** A comparison of MR-VAEs and $\beta$-VAEs (with and without KL annealing). $\beta$-VAEs were trained multiple times on several KL weighting $\beta$ to construct the rate-distortion curve. Across all tasks, MR-VAEs achieve a competitive rate-distortion curve with $\beta$-VAEs on a single run of training.

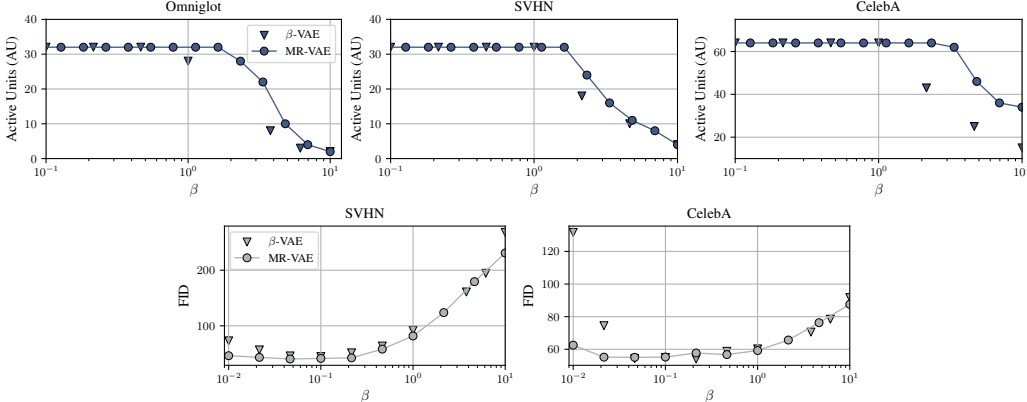

**Figure 6:** A comparison of independently trained $\beta$-VAEs (with KL annealing) and MR-VAEs on proxy metrics such as **(top)** active units (AU) and **(bottom)** Fréchet inception distance (FID). Note that a lower FID is better.

As MR-VAEs directly learn the optimal parameters corresponding to some $\beta$ in a single training run, we can use them to analyze the dependency between the KL weight $\beta$ and various proxy metrics with a single training run. To demonstrate this capability of MR-VAEs, we compute the Fréchet Inception Distance (FID) (Heusel et al., 2017) for natural images and active units (AU) (Burda et al., 2016) in Figure 6. Notably, by using a shared structure for various KL weights, we observed that MR-VAEs are more resistant to dead latent variables with high KL weights and generate more realistic images with low KL weights.

We further show samples from both $\beta$-VAEs and MR-VAEs on SVHN and CelebA datasets which used ResNet-based encoders and decoders in Figure 7. One interesting property is that MR-VAEs have consistency among the generated images for the same sampled latent since base VAE weights are shared for different $\beta$. A practitioner can select the desired model for their task or trade-off between compression and generation quality by feeding different $\beta$s into the hypernetwork at inference time.

### 5.3 How sensitive are MR-VAEs to hyperparameters?

While the MR-VAE eliminates the need to train the network multiple times with different KL weights and a KL schedule, it introduces two hyperparameters $a$ and $b$ that determine the range to sample

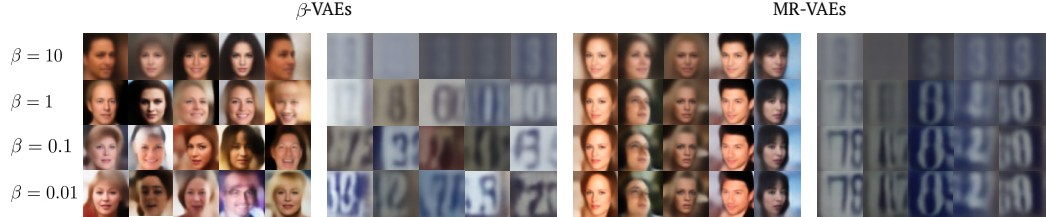

**Figure 7:** Generative samples for $\beta$-VAEs and MR-VAEs on CelebA and SVHN datasets. We use the same latent variables $\mathbf{z}$ sampled from the prior for all columns. For MR-VAEs, there is consistency among the generated images in each column since intermediate weights are shared for different $\beta$. While higher $\beta$ values result in a worse reconstruction loss for both models, MR-VAEs produce images that are of higher quality.

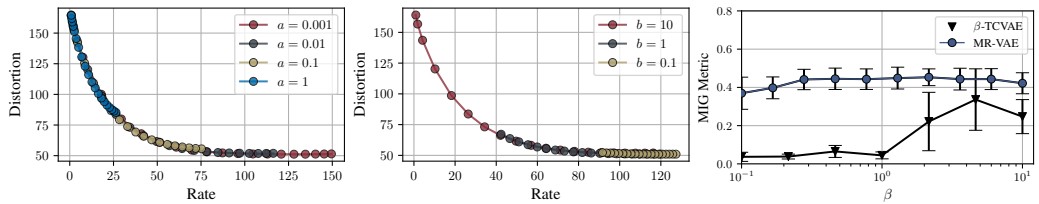

**Figure 8:** **(left and middle)** An ablation studying the effect of the sample range for MR-VAEs. For each figure, we fix one value ($b = 10$ or $a = 0.01$) and change the other. **(right)** A comparison of independently trained $\beta$-TCVAEs and MR-VAEs on the Mutual Information Gap (MIG) metric (the higher value is better)

the KL weight. Here, we show that MR-VAEs are insensitive and robust to the choices of the hyperparameters and can be fixed through various applications. We trained ResNet encoders and decoders on the Omniglot dataset with the same configuration as Section 5.2 and show the test rate-distortion curves in Figure 8. In the left, we fixed $b = 10$ and changed the sample range $a$ in the set $\{0.001, 0.01, 0.1, 1\}$ and in the middle, we fixed $a = 0.01$ and modified the sample range $b$ in the set $\{10, 1, 0.1\}$. While the length of the rate-distortion curves differs with different sample ranges, we observed that the area under the curve is similar across all configurations.

## 5.4 CAN MR-VAEs BE EXTENDED TO OTHER MODELS?

To demonstrate the applicability of MR-VAEs to other VAE models, we trained MR-VAEs on the $\beta$-TCVAE objective. The weight in $\beta$-TCVAE balances the reconstruction error and the total correlation instead of the reconstruction error and the KL divergence term.

We trained MR-VAEs composed of MLP encoders and decoders on the dSprites dataset, following the set-up from Chen et al. (2018). Since we are interested in the disentanglement ability of the model, we sampled the weighting $\beta$ between 0.1 and 10. We compare $\beta$-TCVAEs and MR-VAEs (trained with the $\beta$-TCVAE objective) by examining their performance on the Mutual Information Gap (MIG) in Figure 8. We observed that MR-VAEs are more robust to the choice of the weighting term and achieve competitive final results as the baseline $\beta$-TCVAEs without having to train multiple models.

## 6 CONCLUSION

In this work, we introduce Multi-Rate VAE (MR-VAE), an approach for learning multiple VAEs with different KL weights in a single training run. Our method eliminates the need for extensive hyperparameter tuning on the KL weight while only introducing a small memory and computational overhead. The key idea is to directly learn a map from KL weights to optimal encoder and decoder parameters using a hypernetwork. On various tasks and architectures, we show that MR-VAEs generate competitive or even better rate-distortion curves compared to the baseline of retraining multiple VAEs with various values of $\beta$. Moreover, our framework is general and straightforward to extend to various existing VAE architectures.

## ACKNOWLEDGEMENTS

We would like to thank James Lucas, Rob Brekelmans, Xuchan Bao, Nikita Dhawan, Cem Anil, and Silviu Pitis for their valuable feedback on this paper. We would also like to thank Sicong Huang and many other colleagues for their helpful discussions throughout this research. Resources used in this research were provided, in part, by the Province of Ontario, the Government of Canada through CIFAR, and companies sponsoring the Vector Institute (`www.vectorinstitute.ai/partners`).

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
