# OpenReview forum: "Multi-Rate VAE: Train Once, Get the Full Rate-Distortion Curve"
_ICLR.cc/2023/Conference — ICLR 2023 notable top 5%_

### Official Review · Reviewer_MZus · 2022-10-24

**Confidence:** 4
**Correctness:** 3
**Technical Novelty And Significance:** 4
**Empirical Novelty And Significance:** 4
**Recommendation:** 8

**Clarity, Quality, Novelty And Reproducibility:**

The paper is well organised and clearly written in most parts. However, the technical material given in the supplementary material is indispensable for understanding the approach. Even considering this, there are quite some open questions remaining (see above).

**Strength And Weaknesses:**

Paper strengths:
The authors start from simple log-linear Gaussian VAEs and prove theoretically that a simple affine transform of the encoder/decoder weights parametrised by $\beta$ provides the dependence of the optimal wights on the hyper-parameter $\beta$. They conjecture that the hyper-network for complex, non-linear Gaussian VAEs is obtained by applying such affine transforms of the base weights in each layer of the encoder/decoder pair. The experiments compare achievable data likelihoods for VAEs (with  fixed architecture), when learned with fixed betas (aka response curve), with the one obtained from learning the hypernetwork. It is striking to see how good the corresponding rate curves are matched by the hypernetwork for different VAE variants.

Paper weaknesses:
There are a few conceptual choices and issues which remain unclear for me:
I was not able to fully follow the proofs given in the supplementary material for the log-linear Gaussian VAEs:
- Step from (18) to (19,20): Taking the expectation over the data and maximising it w.r.t. to $\mu$ will give a $\mu_{MLE}$ that depends on the other model parameters. It remains unclear how to derive the gradients of the objective w.r.t. the latter.
- Theorem 1: It remains unclear to me, why it is necessary to represent the parameter matrices of the model as products of
matrices? Why are then the affine transforms applied directly to the VAE weights for the non-linear models in (8,9)?

It remains unclear to me, why applying the affine transform in all layers of the encoder/decoder of non-linear VAEs is sufficient for obtaining the required hypernetwork. Is this may be related to the known rescaling properties of ReLU networks? Would this work also for other non-linearities?. And more general, can this approach be used also for non-Gaussian VAEs with discrete latent variables?

**Summary Of The Paper:**

The paper considers $\beta$-VAEs and treats the weight $\beta$ of the KL-divergence term as a hyper-parameter. The authors develop a hypernetwork model which learns the optimal VAE parameters (i.e. weights of the encoder and decoder) simultaneously for all $\beta$ values within a pre-specified range. The approach is experimentally tested on different datasets like MNIST, SVHN and others as well as for different VAE models like standard VAEs and hierarchical VAEs.

**Summary Of The Review:**

The paper proposes an interesting hypernetwork approach for estimating the optimal weights of $\beta$-VAEs simultaneously for all $\beta$ values within a pre-specified range for given encoder/decoder architectures. The technical part (especially in  the supplementary material) is sometimes hard to follow and there are a few questions and issues remaining open. My current recommendation is a lower bound. The final recommendation will depend on the clarification of these questions in the discussion phase.

After rebuttal discussion: The clarifications provided by the authors answered almost all my questions and are convincing. I am raising my recommendation to 8.

---

> ### Author Response · Authors · 2022-11-12
> **Reply to Reviewer MZus**
>
> Thank you for your feedback and help in improving the paper. We will address your concerns and questions below.
>
> > Q. Theorem 1: It remains unclear to me, why it is necessary to represent the parameter matrices of the model as products of matrices? Why are then the affine transforms applied directly to the VAE weights for the non-linear models in (8,9)?
>
> The product of two matrices in Theorem 1 is necessary to match the functional form of linear VAE response functions. As a motivating example, the derivation in Appendix A.2 highlights that the optimal parameters of the encoder can be written in a form of $\mathbf{A} \mathbf{D}(\beta) \mathbf{B}$, where $\mathbf{D}(\beta)$ formulates a diagonal matrix conditioned on $\beta$. This diagonal matrix can be seen as the scaling of activation after the first layer (or equivalently, the first matrix in the product). Hence, we use two MR-VAE layers to guarantee recovery of the optimal solution.
>
> While the exact response function for deep VAEs is not straightforward to approximate, we use the analysis of linear VAE as a motivation. In Appendix D, we further show that a simple affine transformation obtains similar performance to more expressive architectures such as 2-layer MLP and is sufficient to learn the full rate-distortion curve. We thus use an affine transformation in all our experiments.
>
> > Q. Step from (18) to (19,20): Taking the expectation over the data and maximizing it w.r.t. $\mu$ will give a $\mu_{\text{MLE}}$ that depends on the other model parameters. It remains unclear how to derive the gradients of the objective w.r.t. the latter.
>
> Our linear VAE formulation is equivalent to that presented in [1]. As in [1], we utilize the fact that the optimal parameters for the encoder and decoder coincide with the optimal parameters of the probabilistic PCA (pPCA). Hence, $\mu_{\text{MLE}}$ is the mean of the data and we assume that $\mu$ is fixed throughout the derivation. We apologize for the confusion and will clarify this connection in our updated manuscript.
>
> > Q. It remains unclear to me, why applying the affine transform in all layers of the encoder/decoder of non-linear VAEs is sufficient for obtaining the required hypernetwork. Is this may be related to the known rescaling properties of ReLU networks? Would this work also for other non-linearities?
>
> Previous works (e.g., [2]) highlight that multiplicative interactions (as done in MR-VAE) enrich the representable function classes of neural networks and are successful in many cases where multiple pieces of information need to be combined. We are not completely sure what you mean by the connection to the rescaling properties of ReLU networks. In Section 5, we show that MR-VAEs are compatible with various activation functions; we explored ReLU, ELU, Swish, and Tanh activation functions. Does our response address your question?
>
> > Q. More generally, can this approach be used also for non-Gaussian VAEs with discrete latent variables?
>
> While our analysis and experiments are limited to Gaussian VAEs, it is straightforward to apply MR-VAE methodology to various VAE architectures such as VQ-VAE [3], which we leave for future work. As an example, we can construct the hypernetwork formulation conditioned on $\lambda$ [4] to learn a tradeoff between reconstruction loss and codebook loss + commitment loss in a single training run.
>
> Please let us know if our response addresses your concerns; we are happy to make additional clarifications.
>
> [1] Lucas, J., Tucker, G., Grosse, R. B., & Norouzi, M. (2019). Don't blame the elbo! a linear vae perspective on posterior collapse. Advances in Neural Information Processing Systems, 32.
>
> [2] Jayakumar, S. M., Czarnecki, W. M., Menick, J., Schwarz, J., Rae, J., Osindero, S., ... & Pascanu, R. (2020). Multiplicative interactions and where to find them.
>
> [3] Van Den Oord, A., & Vinyals, O. (2017). Neural discrete representation learning. Advances in neural information processing systems, 30.
>
> [4] Wu, H., & Flierl, M. (2019, November). Learning product codebooks using vector-quantized autoencoders for image retrieval. In 2019 IEEE Global Conference on Signal and Information Processing (GlobalSIP) (pp. 1-5). IEEE.

---

> > ### Comment · Reviewer_MZus · 2022-11-18
> > **Response**
> >
> > Thank you for the detailed clarifications and answers to my questions. I will raise my recommendation.

---

### Official Review · Reviewer_47TW · 2022-10-24

**Confidence:** 4
**Correctness:** 4
**Technical Novelty And Significance:** 3
**Empirical Novelty And Significance:** 3
**Recommendation:** 8

**Clarity, Quality, Novelty And Reproducibility:**

**Clarify & Quality**
The paper is very well-written, and the quality of the exposition is high.

**Novelty**
While the question studied in the paper is not very novel, the paper provides a clean algorithm with good justification that also works well in practice.  The analysis of optimal response functions on linear VAEs was insightful.

**Strength And Weaknesses:**

**Strengths**
* Very well-written paper with a clear motivation.
* Strong empirical results on diverse datasets.  In particular, the paper addresses the important questions that naturally arise: (1) does MR-VAE introduce performance degradation?  (2) does MR-VAE generalize to larger, complicated architectures?  (3) how sensitive is MR-VAE to the choice of its (only) additional hyperparameter $(a, b)$?  The experiments convincingly and positively answer all these questions.

**Weaknesses**
* I did not find any particular weakness.

**Summary Of The Paper:**

This paper proposes MR-VAE, an algorithm to train a VAE that corresponds to an interval of $\beta$ values in a single training run.  The proposed method trains a hypernetwork that modifies the weights of a regular VAE at inference time for a given value of $\beta$, thus eliminating the need to retrain a VAE for each value of $\beta$.  The choice of hyperparameters for the hypernetwork is based on the analysis on linear VAEs, and the empirical results demonstrate that the method generalizes well to modern VAE architectures.

**Summary Of The Review:**

This paper provides an answer to a somewhat obvious but practically important question of whether it's possible to train a VAE that can operate on a continuum of rate-distortion points.  The presentation is clear, with clear experimental results to back up the usability (i.e. R-D performance, computational/memory cost, stability to hyperparameters).  I believe this is a valuable contribution to the field.

---

> ### Author Response · Authors · 2022-11-12
> **Reply to Reviewer 47TW**
>
> We are pleased that the reviewer found our work to be very well-written with strong empirical results. Please let us know if you have any additional comments.

---

### Official Review · Reviewer_fEL1 · 2022-10-30

**Confidence:** 4
**Clarity, Quality, Novelty And Reproducibility:** The paper is well written, and the id…
**Correctness:** 4
**Technical Novelty And Significance:** 4
**Empirical Novelty And Significance:** 4
**Recommendation:** 8

**Strength And Weaknesses:**

Strengths:
1. The proposed hyper-network construction seems to be theoretically principled. This is because for linear VAEs, the paper provides a theorem that proves that indeed the optimized hyper-network can output parameters for the optimal linear $\beta$-VAE.
2. The hyper-network is of practical importance since it bypasses the need to retrain $\beta$-VAEs for different values of $\beta$, while fine tuning for optimal $\beta$.
3. The hyper-network has similar size and training time as the $\beta$-VAE it outputs.
4. The results have been empirically verified on MNIST, OMNIGLOT, CIFAR10 and CelebA datasets, where it is shown that the rate-distortion curve created using the the networks computed from this hyper-network matches the one that is obtained by repeatedly training $\beta$-VAEs for different values of $\beta$.

**Summary Of The Paper:**

This paper introduces Multi-Rate VAE, which is a hyper-network that outputs the parameters of a $\beta$-VAE network, when given a value of $\beta$. The rate-distortion curve created using the the networks computed from this hyper-network matches the one that is obtained by repeatedly training $\beta$-VAEs for different values of $\beta$. This hyper-network also has similar number of parameters as that of the $\beta$-VAE network it outputs, and has similar training time. The construction of the hyper-network is theoretically principled, because the paper proves that the hyper-network can output the optimal $\beta$-VAE when the VAE is linear.

**Summary Of The Review:**

I think the idea of hyper-networks outputting $\beta$-VAE for different values of $\beta$ without the need for retraining from scratch for each $\beta$, is of practical importance. Furthermore, the proposed framework is both theoretically principled and empirically verified to work.

---

> ### Author Response · Authors · 2022-11-12
> **Reply to Reviewer fEL1**
>
> We appreciate the positive review. Please let us know if you have any additional comments.

---

### Author Response · Authors · 2022-11-12
**General Response to All Reviewers**

Dear Reviewers,

We thank all reviewers for their detailed reviews and comments! We are pleased that the reviewers identified our work to be well-written and easy to read (fEL1, 47TW, MZus), interesting and novel (fEL1, MZus), technically solid (fEL1, 47TW), and the experiments to be comprehensive (fEL1, 47TW, MZus).

We will address the remaining concerns and questions in an individual response. We thank the reviewers again for their time.

---

### Public Comment · ~Matthias_Seeger1 · 2023-02-07
**Compare against bilevel optimization techniques?**

I feel that this paper misses a comparison and mentioning of bilevel optimization methods as pretty obvious baselines to compare against.
Since beta is just a single scalar parameter, this should be quite effective.

Luca Franceschi, Michele Donini, Paolo Frasconi, and Massimiliano Pontil. Forward and reverse gradient-based hyperparameter optimization. ICML 2017

Riccardo Grazzi, Luca Franceschi, Massimiliano Pontil, and Saverio Salzo. On the iteration com- plexity of hypergradient computation. ICML 2020.

https://arxiv.org/abs/2212.14032 even has one of the authors here on the paper as well.

---

> ### Author Response · Authors · 2023-05-02
> **Thank you for your feedback and clarification on our paper's objective**
>
> Hello,
>
> We sincerely appreciate your valuable comment and suggestion. We would like to emphasize that our paper's primary goal is to explore the Pareto frontier of various objectives, including qualitative and non-differentiable metrics (such as sampled images, FID, and AU scores), by adjusting values of $\beta$. Nonetheless, we agree that bilevel optimization can be effectively employed with our response functions if the aim is to find the best $\beta$ value. In light of your input, we will incorporate a more extensive discussion of bilevel optimization methods in our updated manuscript. Thank you again for your constructive feedback.

---

### Public Comment · ~Alexey_Dosovitskiy1 · 2023-04-30
**related work**

Hi,

Nice work, I feel like our paper from a couple years ago is pretty related and should perhaps be mentioned and compared to  https://openreview.net/forum?id=HyxY6JHKwr (we trained beta-VAE with all values of beta at once as one of experiments in the paper)

---

> ### Author Response · Authors · 2023-05-02
> **Thank you for your input**
>
> Hello,
>
> We greatly appreciate your recommendation. The paper you've brought to our attention indeed appears to be highly relevant. We will incorporate it into the forthcoming revision of our manuscript.

---

### Decision · Program_Chairs · 2023-01-20

**Decision:**

Accept: notable-top-5%

**Justification For Why Not Higher Score:**

N/A

**Justification For Why Not Lower Score:**

The paper had very strong reviews, with no concerns. At the same time, the paper is highly relevant to a large part of the community.

**Metareview: Summary, Strengths And Weaknesses:**

This paper proposes a method to learn a hypernetwork for Beta VAEs - a network that can output the parameters of a VAE for all possible values of beta. The paper received very strong reviews which praised its very well-written, high-quality exposition, its theoretical justification and its strong experimental results. While there were a very small number of concerns by the reviewers, the authors have adequately addressed all of them, with the paper having no outstanding concerns from the reviewers. This paper is further of great interest to the community as Beta VAEs are widely used and the paper provides a principled approach to get rid of a fundamental difficulty of Beta VAEs, i.e. that one has to retrain the model for all different values of beta. I hence strongly recommend to accept this paper.

**Note From Pc:**

if the above contains the word "oral" or "spotlight" please see: "oral" presentation means -> notable-top-5% and "spotlight" means -> notable-top-25%. As stated in our emails, we are disassociating presentation type from AC recommendations